

# Differentiating fire regimes and their biophysical drivers in Central Portugal

Rafaello Bergonse[1], Sandra Oliveira[1], José Luís Zêzere[1], Francisco Moreira[2], Paulo Flores Ribeiro[3], Miguel Leal[3], José Manuel Lima e Santos[3]

[1]Centre for Geographical Studies, Institute of Geography and Spatial Planning and Associate Laboratory TERRA, Universidade de Lisboa. Rua Branca Edmée Marques, Cidade Universitária, 1600-276 Lisbon, Portugal

[2]CIBIO – Research Centre in Biodiversity and Genetic Resources, Universidade do Porto. Campus de Vairão. Rua Padre Armando Quintas, nº 7, 4485-661 Vairão, Portugal

[3]Forest Research Centre, Instituto Superior de Agronomia and Associate Laboratory TERRA, Universidade de Lisboa. Edifício Prof. Azevedo Gomes, Instituto Superior de Agronomia, Tapada da Ajuda, 1349-017 Lisboa

*Correspondence to*: Rafaello Bergonse (rafaellobergonse@campus.ul.pt)

**Abstract.** The spatial and temporal patterns of wildfires and their effects over a given area can be described using the concept of fire regime. Here, we characterize fire regimes Central Portugal and investigate the degree to which the differences between regimes are influenced by a set of biophysical drivers. Using civil parishes as units of analysis, we employ three complementary parameters to describe the fire regime over a reference period of 44 years (1975-2018): cumulative percentage of parish area burned, Gini concentration index of burned area over time, and area-weighted total number of wildfires. Cluster analysis is used to aggregate parishes into groups with similar fire regime based on these parameters. A classification tree model is then used to assess the capacity of a set of potential biophysical drivers to discriminate between the different parish groups. Drivers included slope, summer temperature and spring rainfall, land use/land cover (LULC) type and patch fragmentation, and net primary productivity. Results allowed to distinguish four types of fire regime and show that these can be significantly differentiated using the biophysical drivers, of which LULC, slope and spring rainfall are the most important. Among LULC classes, shrubland and herbaceous vegetation play the foremost role, followed by agriculture. Our results highlight the importance of vegetation type, availability, and rate of regeneration, as well as that of topography, in influencing fire regimes in the study area, while suggesting that these regimes should be subject to specific wildfire prevention and mitigation policies.

**Keywords:** Fire regime, Biophysical drivers, Machine learning, Classification and regression trees, Central Portugal



## 1. Introduction

The characteristics of wildfire activity, such as frequency, intensity, seasonality, and type of fuels consumed, determine the fire regime (Pausas & Keeley, 2014), which can be defined as the spatial and temporal patterns of fires and their effects within a given area and period of time (Oddi, 2018). Fire regimes result from the interactions of fire with different biophysical, climatic, and anthropogenic factors. From a hazard management perspective, it is essential to understand these interactions due to the human, material, and environmental damage caused by wildfires. Numerous studies have focused on the influence of biophysical factors such as climate, topography, land use/land cover (LULC), as well as social factors as demographics and road density, over properties of the fire regime across different periods. For example, Oliveira and Zêzere (2020) used a local-scale approach to explore the relations between biophysical and social factors and wildfire incidence in Portugal with reference to an 8-year period, showing both LULC and socioeconomic conditions to be important drivers of burned areas. Also for Portugal, Fernandes et al. (2019) focused on the consequences of expanding Eucalyptus stands on different wildfire properties over 38 years. González et al (2018) centred their attention on the effect of a five-year period of dry atmospheric conditions over several fire regime properties in central and south-central Chile. Curt et al. (2016) focused on the biophysical and anthropic causes of ignitions in southeast France over a 41-year period, showing that the human role in originating ignitions influences such diverse fire regime properties as potential wildfire size, location, and timing of occurrence within each year. On a greatly contrasting timescale, Connor et al. (2019) compared sedimentary charcoal data (as a proxy for fire occurrence) and pollen records for a period extending to most of the Holocene, drawing attention to the long-scale effect of human populations over fire regime, and ultimately on vegetation type and diversity in Mediterranean Iberia.

Portugal is one of the southern European countries with more wildfire damage. Its average annual burned area between 1980 and 2019 amounted to 115 024 ha, a value surpassed only by Spain (San-Miguel-Ayanz et al., 2020). Most ignitions and burned area in Portugal are concentrated north of the Tagus River, where irregular topography is combined with forests and semi-natural land cover. The southern half of the country, except southernmost Algarve, is dominated by lowlands associated with agriculture and agro-forestry, showing a markedly lower wildfire incidence (Nunes et al., 2016; Oliveira et al., 2020; Tonini et al., 2017). The largest burned areas are concentrated in the central sector of the country, which is dominated by forest and shrubland and has been the subject of several studies (e.g., Catry et al., 2010; Maia et al., 2012). This area was the most affected by the extreme wildfires that took place in 2017 (Benali et al., 2021).

Recently, Bergonse et al. (2022) described fire regime over a 44-year study period in Central Portugal using three complementary variables: cumulative percentage of parish area burned, area-weighted total number of wildfires, and the Gini concentration index of burned area over time. They then quantified the influence of a set of 12 biophysical variables over each of these fire regime descriptors using ordinal logistic regression. Although the fire regime was assumed to be the same throughout the entire study area, the authors noted that contrasting spatial distributions among the three fire regime descriptors suggested the existence of at least three distinct fire regimes. The present work draws on these results, with the main goal of identifying and characterizing distinct fire regimes in Central Portugal. In addition, this research intends to investigate the role





of biophysical conditions in distinguishing these fire regimes. Understanding how the interactions between fire regimes and
biophysical factors affect wildfire hazard levels is paramount to adapt fire management strategies.

**2 Data and Methods**

**2.1 Study area**

The study area occupies 28 199 km$^2$ in central mainland Portugal, corresponding to the whole of the NUTS II *Centro* (Fig. 1).
This region is marked by a high variability regarding wildfire hazard and its control factors (Oliveira et al., 2020). Elevation
increases inland in an eastern direction, with the Central Mountain Range (*Cordilheira Central*) traversing the study area in a
SW-NE direction. Annual rainfall also varies significantly, from 600 mm in the extreme NE to 2400-2800 mm in the highest
sectors of the Central Mountain Range (Brito, 2005). Regarding LULC distribution, coniferous and eucalyptus forests
dominate the centre of the study area in a broad S-N swath. A second extensive, pine-dominated area occurs along the shoreline.
The highest sector of the Central Mountain Range is dominated by shrubland and sparsely vegetated or unvegetated terrain,
with the E and SW sectors of the study area being characterized by shrubland and agro-forestry/agriculture. Urban areas are
mostly concentrated near to the shoreline, where the highest population densities are found (Brito, 2005).
From a structural perspective, about half the region (49.7%) is classified in the High and Very High wildfire hazard classes
(Oliveira et al., 2020), with these classes being mostly concentrated in the central and northeastern sectors of the study area.
Smaller and localized spots of high wildfire hazard are also found in the W sector.
The spatial units of analysis were the 972 civil parishes comprised within the study area (Fig.1). These constitute the smallest
administrative units in Portugal with management responsibilities over its territory (Oliveira & Zêzere, 2020). Their areas vary
between 1.98 km$^2$ and 373.50 km$^2$. Parish boundaries were extracted from the official administrative map of Portugal (CAOP),
(Portuguese Directorate-General of the Territory, DGT).



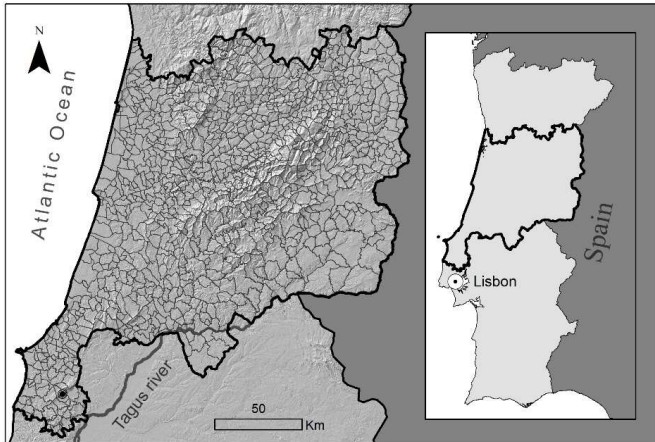


**Figure 1: Boundaries and position of the study area within mainland Portugal (NUTS II *Centro*). Parish limits are also shown.**

### 2.2 Data collection and pre-processing

### 2.2.1 Fire regime parameters

Fire regime characteristics were analysed using three descriptors, obtained for a reference period of 44 years (1975-2018), the longest time-series available. These data were all obtained from the burned area vector maps produced annually by the National Forests Service (ICNF). Following prior research developed by Bergonse et al. (2022), the cumulative percentage of parish area burned (CPAB) was used to measure the propensity of each parish to burn extensively over time. Area-weighted wildfire frequency (AWWF) was calculated as the total number of wildfires recorded within the parish over the study period, divided

by the parish area (in km$^2$) in order to avoid scale effects because of contrasting parish sizes.

The Gini Concentration Index (GCI) of burned area over time was adopted as an indicator of the temporal concentration of wildfire damage. It corresponds to the Gini Concentration Index, applied to the annual burned areas of each parish over the 44 years. The GCI corresponds to the Gini coefficient when expressed in percentage. The Gini coefficient $G$ can be formulated as (Brown, 1994):

$$G = 1 - \sum_{i=0}^{K-1}(X_{i+1} - X_i)(Y_{i+1} + Y_i) \qquad (1)$$

Where $k$ is the total number of years (44), $X$ is the cumulative percentage of years associated with the i[th] year, and $Y$ is the cumulative percentage of burned area associated with the same year. Varying in a scale between 0 and 100, the GCI allows to differentiate parishes where most burned area is concentrated in a small number of years (high values), from those where the burned area is more regularly distributed over time (low values). However, the GCI does not quantify the magnitude of the

concentrated or distributed burned area, being complementary to CPAB.



### 2.2.2 Potential fire regime drivers

Prior research developed for the study area indicated an association between fire regime parameters and particular biophysical conditions (Bergonse et al., 2022). A set of 12 biophysical variables was adopted (Table 1), which were found to be significantly associated with the three fire regime parameters under analysis (Bergonse et al., 2022), although no differentiation has been tested regarding their spatial distribution and aggregation.

Topography was expressed by slope (80th percentile, in degrees), which can be expected to promote flame propagation (Carmo et al., 2011; Gralewicz et al., 2012; Leuenberger et al., 2018). It was obtained from the 25 m pixel European Environmental Agency's Digital Surface Model (https://www.eea.europa.eu/data-and-maps/data/copernicus-land-monitoring-service-eu-dem).

The role of climate was expressed using two variables. Cumulative rainfall during the spring months (April-June) (RFAJ) was adopted to represent the potential effect of spring rainfall over the flammability of existing fuel during spring, as well as on the production of fuel potentially available to burn later in the year. Rainfall off the summer season was observed by Oliveira et al. (2012) to be a positive influence over wildfire occurrence in Southern Europe, suggesting a positive effect of spring rainfall on fuel accumulation. RFAJ was calculated from monthly rainfall data obtained from the Worldclim database (1970-2000), available at https://www.worldclim.org (Fick & Hijmans, 2017), in the form of raster maps of approximately 30 seconds (about 1 km resolution), which were resampled to a 25 m pixel.

Mean monthly temperature during the summer months (Jul-Sep) (TPJS) was used to represent the potential role of air temperature over fuel flammability during the summer (Aldersley et al., 2011; Ventura & Vasconcelos, 2006). It was calculated from mean monthly temperature raster maps (30-second resolution) extracted from the Worldclim database (reference period 1970-2000), resampled to a 25 m pixel.

Land use/Land cover (LULC) was obtained from the official Land Use/Land Cover maps (*Carta de Uso e Ocupação do Solo*) for the available years (1990, 1995, 2007, 2010, 2015 and 2018), produced by the Portuguese General-Directorate of the Territory. Seven class aggregations were used, representing areas with similar types of vegetation and land occupation. All were expressed as percentage of the parish area. AGR combined all agricultural land uses including orchards, vineyards, olive groves, permanent pastures, temporary dryland and irrigated cultures, temporary cultures and/or pastures associated with permanent cultures, as well as complex land parcel and cultivation systems and rice paddies. SHR included areas occupied by spontaneous herbaceous vegetation and shrubland. The latter is the most fire-prone LULC type in Portugal (Carmo et al., 2011; Marques et al., 2011; Moreira et al., 2009; M. C. S. Nunes et al., 2005), as well as in Mediterranean-type areas in general (Curt et al., 2013; Oliveira et al., 2012).

The remaining five aggregations are forest-based. According to the technical specification of the LULC cartography used, the classification "forest" requires the presence of trees of at least 5 m height that cover a minimum of 30% of the ground surface (Caetano et al., 2018). OAK included holm-oak (*quercus rotundifolea*) and cork oak (*quercus suber*). EUC included eucalyptus forests (mostly *Eucalyptus globulus*). CON included forests of coniferous species other than stone or maritime pine. These



include other *Pinus* species as well as *Larix*, *Picea* or *Abies* species. BRD included forests of broadleaves other than holm oak, cork oak and eucalyptus. It includes species such as Pyrenean oak (*Quercus pyrenaica*), chestnut oak (*Castanea sativa*), and European oak (*Quercus robur*), as well as species of *Salix*, *Populus* or *Platanus*. INV included all forests of invasive species (e.g., *Ailanthus altissima*, *Acacia dealbata*).

The percentage of each LULC class for each parish was calculated as the mean between the values corresponding to the six

existing LULC maps encompassed by the study period (1990, 1995, 2007, 2010, 2015, 2018), weighted by the number of years during which each LULC map was valid.

Net Primary Productivity (NPP) was employed as a proxy for biomass and therefore fuel availability (Pausas & Ribeiro, 2013). It was calculated from annual maps of NPP (in $KgC/m^2$) between 2000 and 2014 (the available period) obtained from NASA's Earth Science Data Systems database (https://lpdaac.usgs.gov/products/mod17a3hgfv006/) and resampled from the original

500 m pixels to 25 m pixels. Mean annual values were calculated from the 15 available years. Finally, the mean value among the pixels in each parish was calculated.

LULC patch fragmentation has a well-known influence over the capacity of wildfire to propagate efficiently (Curt et al., 2013; Fernandes et al., 2016; Gralewicz et al., 2012; Turner & Romme, 1994). Following Bergonse et al. (2022), we calculated the fragmentation of forest patches by merging all forest patches into a single polygon, dividing them into individual unconnected

polygons, and generating the centroid for each of these. The number of centroids contained within each parish was quantified, and then divided by the forest area of the parish (in ha). The final values quantify the mean numbers of disconnected patches per hectare of forest in each parish. As described above for the LULC variables, this procedure was performed for the LULC maps of 1995, 2007, 2010, 2015 and 2018, with the final values being combined as a weighted mean. The 1990 map was not included, due to it having positioning errors (Caetano et al., 2018), likely to influence the results of spatial arrangement-

oriented analyses.

A description of all potential fire regime drivers is shown in Table 1. There are some differences in the period used to characterize the fire regimes (1975-2018, 44 years) and the potential drivers: 1970-2000 (31 years) for all climate variables, 1990-2018 (29 years) for most LULC classes, 1995-2018 (24 years) for the LULC patch fragmentation indicators, and 2000-2014 (15 years) for NPP. These disparities resulted from data availability constraints, and their joint analysis assumes that all

of them are representative of an equivalent long-term perspective.

All variables (fire regime descriptors and potential biophysical drivers) were estimated for the territory of each parish. ArcMap 10.7.1 (ESRI Inc.) was employed for all spatial analysis operations. A 25 m pixel was employed for all raster operations, following the resolution of the topographic data. Variable values were then exported to SPSS 24 (IBM Corp.), which was used for all statistical analyses.


**Table 1 – Description and characteristics of the potential fire regime drivers.**



| Type | Variable code | Variable | Temporal extent | Original spatial Resolution | Units |
|---|---|---|---|---|---|
| Topography | SLO80 | Slope percentile 80 | n.a. | 25 m | ° |
| Climate | RFAJ | Mean cumulative rainfall April-June | 1970-2000 | Approx. 1000m | mm |
| | TPJS | Mean monthly temperature July-September | | | ⁰C |
| Biomass | NPP | Net Primary Productivity | 2000-2014 | 500 m | KgC/m² |
| Land use/Land cover | AGR | % parish area occupied by agriculture | 1990-2018 | Vector data. Minimum mapped area 1 ha | % |
| | OAK | % parish area occupied by holm-oak and cork-oak forests | | | |
| | EUC | % parish area occupied by eucalyptus forests | | | |
| | INV | % parish area occupied by forests of invasive species | 1995-2018 | | |
| | CON | % parish area occupied by forests of coniferous species other than maritime or stone pine | | | |
| | BRD | % parish area occupied by forests of broadleaved species other than holm-oak, cork-oak and eucalyptus | 1990-2018 | | |
| | SHR | % parish area occupied by brushland and spontaneous herbaceous species | | | |
| LULC patch fragmentation | FRAGF | Fragmentation of forest patches | 1995-2018 | | centroids/ha of forest |

n.a. - not applicable

### 2.3 Cluster analysis

Cluster analysis is a multivariate exploratory technique which allows to aggregate subjects, or variables, in mutually exclusive
homogeneous groups, regarding one or more common properties. Calheiros et al. (2020) and Trigo et al. (2016) used it to group spatial administrative units in Iberia based on their monthly normalized burned area. Moreira et al. (2009) aggregated ecological regions within Portugal using each region's wildfire selectivity ratios for different LULC classes. In a contrasting approach, Papagiannaki et al. (2020) employed the same technique to group wildfires regarding their size and associated meteorological conditions, quantified using the Fire Weather Index.

We employed hierarchical cluster analysis to investigate the existence of groups of parishes with similar behaviour regarding the fire regime parameters. Clustering was performed using Ward's method, an agglomerative process which begins with as many clusters as cases, successively agglomerating clusters using the solution that minimizes within-cluster variance (Everitt et al., 2011). Prior to inclusion, the three fire regime parameters were converted into z-scores, to ensure that all have an equal contribution to the final result regardless of contrasting value ranges (Maroco, 2007).






**2.4 Classification tree**

Classification and regression trees are a non-parametric technique developed by Breiman et al. (1984), which progressively divides units of analysis into smaller and smaller groups, designated as nodes, with increasing similarities in the dependent variable within each group, based on critical thresholds in continuous or categorical independent variables (Kelly & Meentemeyer, 2002; Ma, 2018). It presents several advantages of other statistical techniques, such as its capacity to capture complex interactions and nonlinear relationships in the data, its mathematical simplicity, being free from distributional assumptions, and ease of interpretation (De'Ath & Fabricius, 2000; Ma, 2018). It can be subdivided into classification tress and regression trees, whether the dependent variable is categorical or continuous. Both have been applied to wildfires. Classification trees were used by Lozano et al. (2008) to predict the binary condition of burned/unburned in terms of a set of environmental predictors in NW Spain. A similar approach was taken by Jaafari et al. (2018) for the Zagros Mountains in Iran. Regression trees were employed by Aldersley et al. (2011) to assess the effect of different climatic and human variables on burned areas on a global scale. Amatulli et al. (2006) used the technique to model the influence of multiple environmental factors over wildfire density in SE Italy, and Fernandes et al. (2016) applied it to assess the drivers of the size of large fires (>100 ha) for mainland Portugal. Other authors have applied other tree-based techniques, such as random forests, to fire occurrence and susceptibility modelling (Oliveira et al., 2012; Oliveira & Zêzere, 2020; Rodrigues & De La Riva, 2014). Recently, Jain et al. (2020) reviewed the applications of these and other machine learning techniques in wildfire science and management.

SPSS's CRT tool was employed to build a classification tree model with the purpose of assessing the capacity of the 12 biophysical factors to differentiate between the clusters associated with different fire regimes, obtained using cluster analysis (see section 2.3). The values of all factors were converted into z-scores prior to inclusion. The *Gini* criterion was used as measure of impurity (i.e., degree of heterogeneity regarding the values of the dependent variable) for node splitting. A 10-fold cross-validation procedure was adopted, according to which ten trees are built, each being based on 9/10 of the units of analysis. Each tree is then used to classify the 1/10 of the dataset left out of its construction. The tool produces a final tree, its classification error being the average of the 10 error values obtained during cross-validation.

**3 Results**

**3.1 Cluster analysis**

Out of the total of 972 parishes, 35 (3.6%) never burned during the study period, having therefore no values in any of the fire regime parameters. These were removed from all analyses.

A graphical representation of the distance between clusters associated with solutions ranging between 1 and 25 clusters is shown in Fig. 2. Distances decrease sharply between solutions with up to 3 clusters, decreasing smoothly from this point on. This indicates that a three-cluster solution will incorporate the major fire-regime patterns within the study area, with any larger





number of clusters describing relatively less important nuances. In face of these results, three and four cluster solutions were tested, whose descriptive statistics are shown in Table 2 and illustrated graphically in Fig. 3.


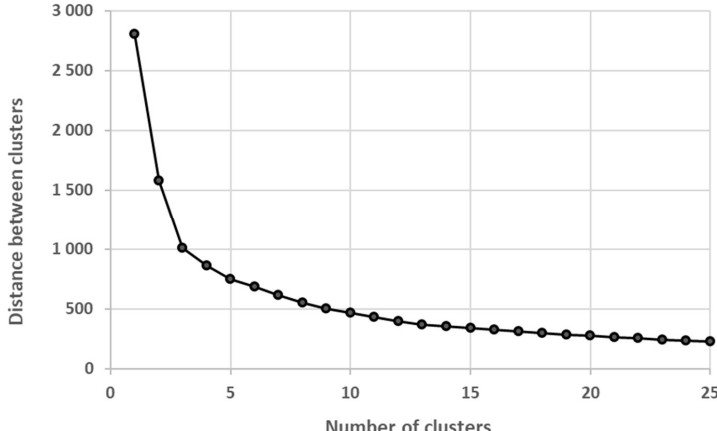

**Figure 2: Distance between clusters throughout successive agglomerations. Values are only shown up to 25 clusters to facilitate visual analysis.**


**Table 2: Descriptive statistics for the values of the three fire regime parameters in each clustering solution. CPAB – Cumulative percentage of area burned; AWWF – Area-weighted wildfire frequency; GCI – Gini Concentration Index. SD – standard deviation.**

|  | Cluster 1 | | Cluster 2 | | Cluster 3 | | Cluster 4 | |
|---|---|---|---|---|---|---|---|---|
| No of parishes | 450 | | 401 | | 86 | | | |
| Variable | Mean | SD | Mean | SD | Mean | SD | | |
| CPAB | 37.6 | 41.0 | 130.1 | 66.6 | 240.7 | 100.0 | | |
| AWWF | 0.4 | 0.3 | 1.18 | 0.56 | 3.33 | 0.96 | | |
| GCI | 94.3 | 2.7 | 84.6 | 5.6 | 74.6 | 6.2 | | |
| No of parishes | 450 | | 299 | | 86 | | 102 | |
| Variable | Mean | SD | Mean | SD | Mean | SD | Mean | SD |
| CPAB | 37.6 | 41.0 | 102.9 | 47.6 | 240.7 | 100.0 | 209.9 | 47.5 |
| AWWF | 0.4 | 0.3 | 1.3 | 0.6 | 3.3 | 1.0 | 0.9 | 0.4 |
| GCI | 94.3 | 2.7 | 83.4 | 5.6 | 74.8 | 6.2 | 87.9 | 3.8 |




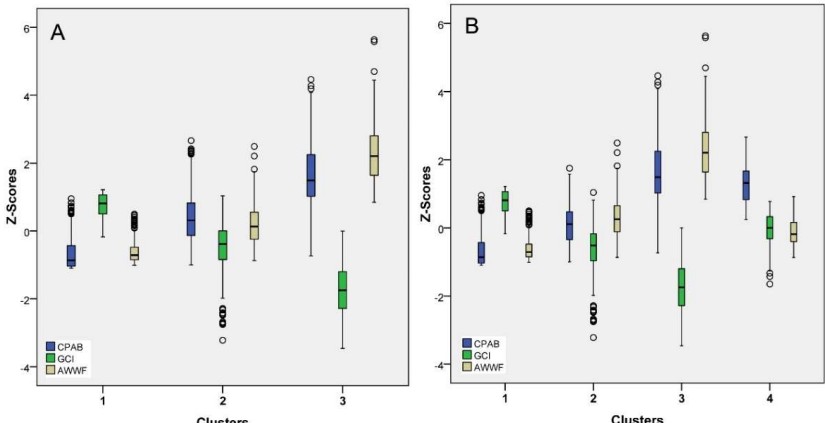

**Figure 3: Boxplots for the values of the three fire regime parameters associated with each clustering solution: 3 clusters (A); 4 clusters (B). Values expressed as z-scores. For each variable, the box includes the 1st and 3rd quartiles as well as the median. The whiskers identify the maximum and minimum values excluding outliers. Outliers (shown as circles) are defined as values between**

**1.5 times and 3 times the interquartile range, respectively above the 3rd quartile and below the 1st quartile.**

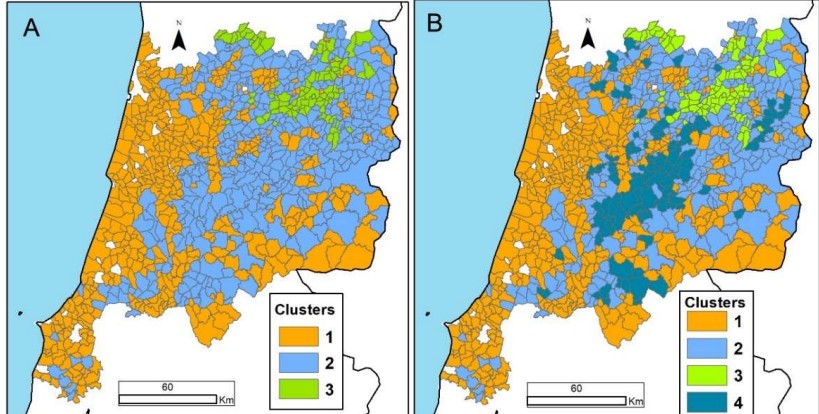

**Figure 4: Spatial distributions of the 3-cluster (A) and 4-cluster (B) solutions.**

Regarding the 3-cluster solution (Fig. 4-A), cluster 1 is characterized by the lowest CPAB, the highest GCI, and the lowest AWWF values within the study area (Table 2 and Fig. 3-A). These values express a fire regime marked by the lowest extension of burned areas and the lowest wildfire frequency within the study area, with the resulting damage being relatively concentrated





over time (corresponding to the highest GCI obtained). Spatially, it occurs mostly along the coastal swath and in the SE extreme of the study area, with some additional parishes occurring in a disperse pattern. It includes 450 parishes (Table 2).

Because it expresses the opposite characteristics, the cluster 3 shows a noteworthy contrast with the first. It has the highest CPAB values found within the study area, as well as the highest AWWF and the lowest GCI (Table 2 and Fig. 3-A). These identify a regime marked by relatively frequent wildfires, which produce very extensive burned areas over time and result in a relative temporal dispersion of the damage. This cluster is the least numerous of the three (86 parishes), occurring exclusively in the NE and the northern limit of the study area.

Cluster 2 occupies an intermediate position between the other two in terms of all three fire regime variables. It shows intermediate tendency for extensive burned area, intermediate wildfire frequency, and intermediate temporal concentration of burned damage (Table 2 and Fig. 3-A). Spatially, it occupies most of the central and eastern portions of the study area, aggregating 401 parishes.

The four-cluster solution results simply from the division of the former cluster 2 into two new clusters, now numbered 2 (with 299 parishes) and 4 (102 parishes) (Table 2 and Fig. 4-B). Cluster 2 in the four-cluster solution is equivalent to the above-described cluster 2 in the three-cluster solution, showing intermediate values between clusters 1 and 3 in all fire regime variables (Fig. 3-B). The new cluster 4, on the other hand, consists of the fraction of parishes of the previous cluster 2 that burn more extensively (higher CPAB) and less frequently (lower AWWF), having thus a greater temporal concentration of damage (higher GCI) (Fig. 3-B). Spatially, cluster 4 is concentrated in the central sector of the study area (Fig. 4-B), with minor parish concentrations in the south and east, and a few dispersed parishes in the northern sector.

Regarding the choice between the two clustering solutions, two points warrant attention. On the one hand, a consideration of the distances between clusters (Fig. 2) indicates that the three-cluster solution expresses, in a more synthetic way, the major differences in fire regime across the study area. On the other hand, cluster 4 shows a clear spatial pattern (Fig. 4-B) and expresses a fire regime that merits attention in terms of wildfire prevention and suppression policies, as it includes the second most extensive burned areas (after cluster 3) (Fig. 3-B). We therefore adopted the four-cluster solution for defining fire regimes (FRs) within the study area, basing all subsequent analyses on this solution.

### 3.2 Classification tree (CT) model

The accuracy of the CT model built using the 12 potential drivers to discriminate between the four FRs is shown in Table 3. The final tree model correctly classified 72.4% of all parishes, with the accuracy being slightly inferior (68.7%) when independently validated using a 10-fold cross-validation process.

**Table 3: Classification accuracy for the final tree model and for the tree models produced in association to the 10-fold cross-validation procedure. FR-specific accuracy values are for the final tree model.**




| Observed | Predicted | | | | % Correct |
|---|---|---|---|---|---|
| | 1 | 2 | 3 | 4 | |
| 1 | 369 | 70 | 1 | 10 | **82.0** |
| 2 | 58 | 209 | 17 | 15 | **69.9** |
| 3 | 0 | 22 | 61 | 3 | **70.9** |
| 4 | 17 | 42 | 4 | 39 | **38.2** |
| | | | **Final tree accuracy** | | **72.4** |
| | | | **Cross-validated accuracy** | | **68.7** |

All 12 biophysical drivers were integrated into the CT model, although with contrasting relative contributions (Fig. 5). The percentage of shrubland and spontaneous herbaceous vegetation (SHR) was the most important factor, closely followed by spring rainfall (RFAJ). Slope (SLO80) and agriculture (AGR) have about half of the importance of SHR and RFAJ, whereas

eucalyptus forests (EUC), broadleaved species other than holm oak, cork oak and eucalyptus (BRD), and net primary productivity have a relative importance between 40% and 50% of SHR. Finally, forest patch fragmentation (FRAGF), summer temperatures (TPJS), forests of invasive species (INV), forests of holm oak and cork oak (OAK) and forests of conifers other than maritime pine and stone pine (CON) have lesser contributions for the model, from a little above 30% of SHR (FRAGF) down to less than 10% (CON).


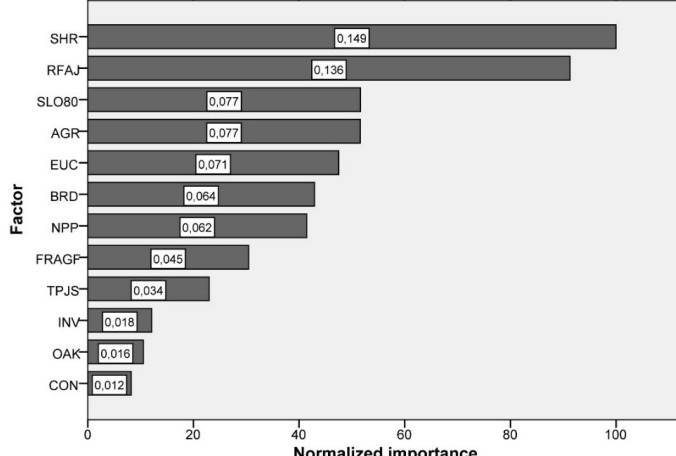

**Figure 5: Importance of each biophysical driver for the classification tree model, shown normalized by the most important driver (SHR).**





To facilitate the interpretation of the role of the different biophysical drivers in discriminating between FRs, boxplots of the
       values of each driver in each of the four FRs are presented in Fig. 6, with the drivers organized by decreasing order of
       importance in the classification tree model.


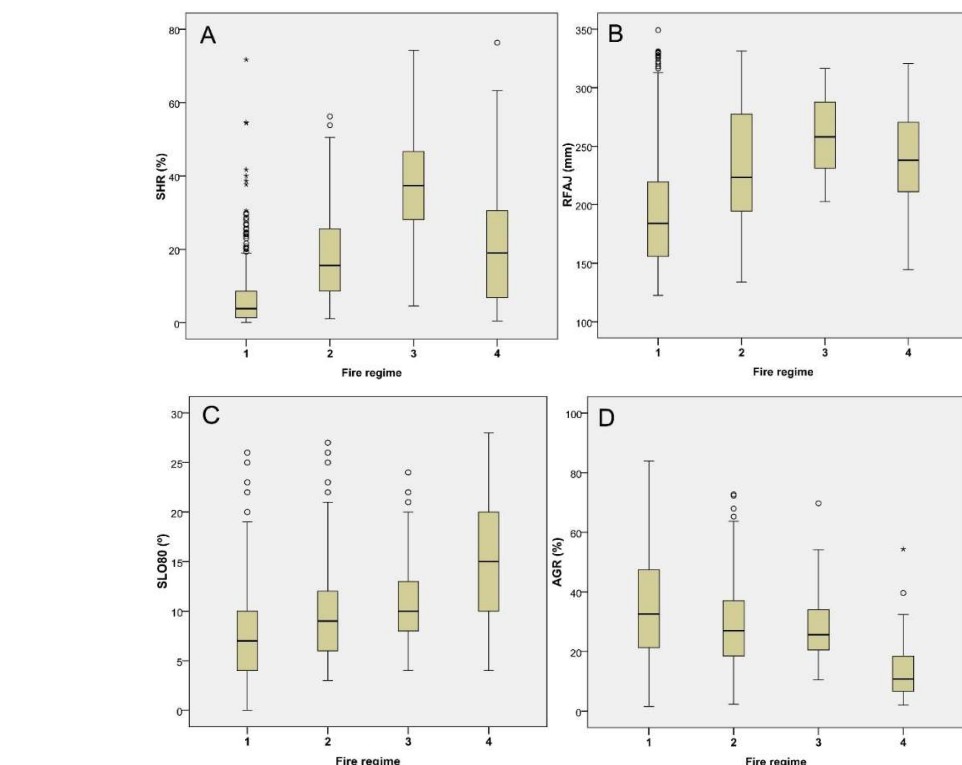

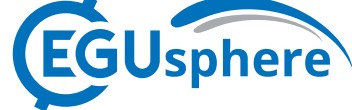

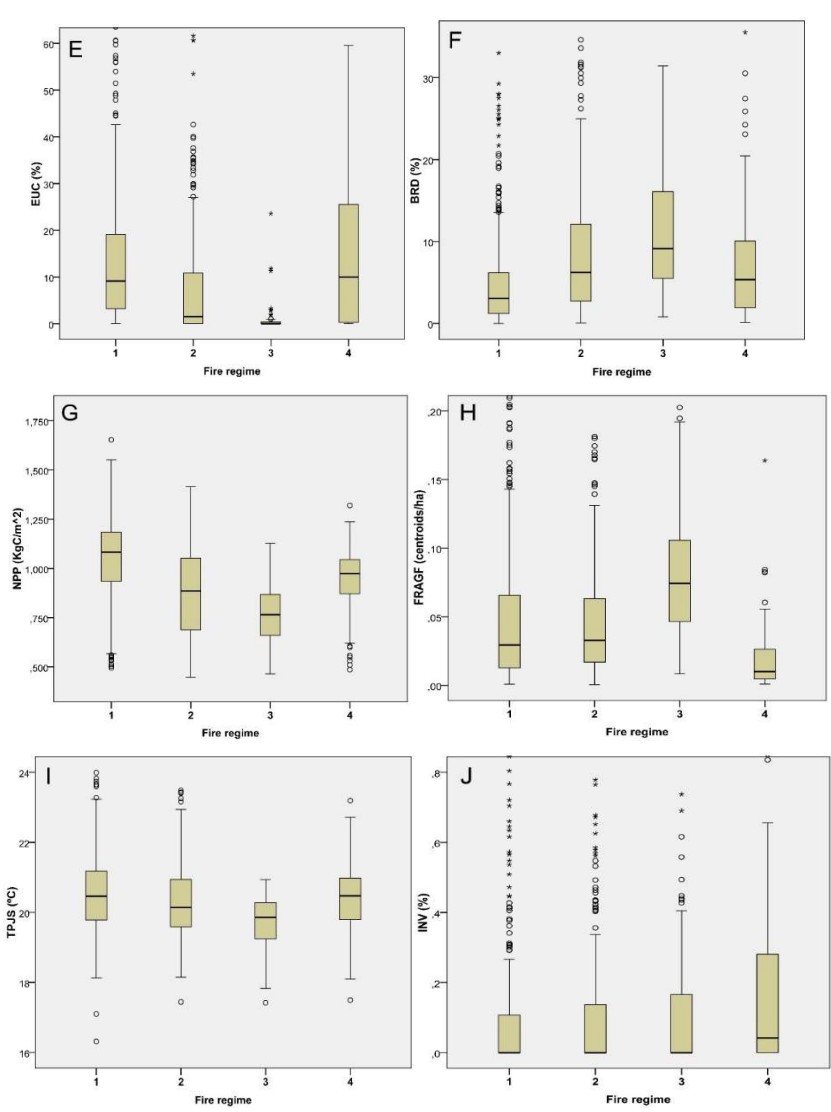



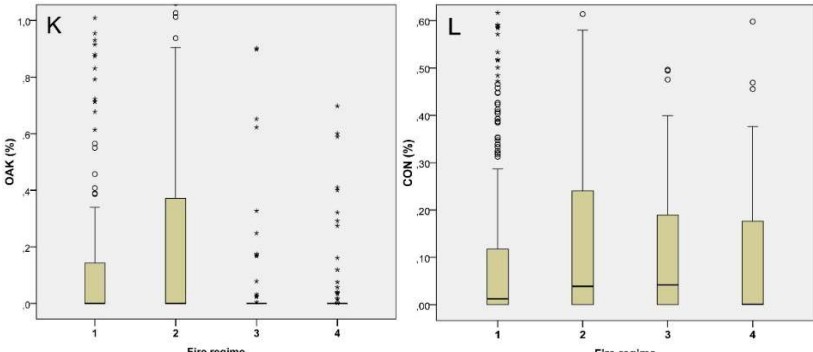


**Figure 6: Boxplots for the values of each biophysical driver in each of the four fire regimes, by order of importance in the classification tree model. A – SHR; B – RFAJ; C – SLO80; D – AGR; E – EUC; F – BRD; G – NPP; H – FRAGF; I – TPJS; J – INV; K – OAK; L – CON. Circles identify potential outliers, defined as situated between 1.5X and 3X the interquartile range below the 1st quartile and above the 3rd quartile. Asterisks identify potential extreme outliers, exceeding 3 times the interquartile range below**

**or above the 1st and the 3rd quartile. The boxplots are ordered by decreasing order of importance of the drivers in the classification tree model.**

FR1 has the lowest values of percentage of parish area occupied by shrubland (SHR; Fig. 6-A), the lowest amount of spring rainfall (RFAJ; Fig. 6-B), the lowest slope inclination (SLO; Fig. 6-C) and the highest percentage of parish area occupied by

agriculture (AGR; Fig. 6-D), as well as the highest Net Productivity Ratio (NPP; Fig. 6-G).

In opposition to FR1, FR3 presents the highest median values of SHR and RFAJ and the lowest NPP. It also has the highest degree of forest patch fragmentation (FRAGF; Fig. 6-H) and the second highest SLO. It has the lowest percentage of eucalyptus forests (EUC) (Fig. 6-E), and the highest incidence of forests of broadleaves other than holm oak, cork oak and eucalyptus (BRD; Fig. 6-F).

FR2 occupies a somewhat intermediate position between FR1 and FR3, as seen by its values in SHR, RFAJ, SLO, AGR, NPP and FRAGF.

FR4 has a relatively high percentage of SHR, high RFAJ, the highest SLO, and the lowest AGR. It also has relatively high EUC and NPP.

**4 Discussion**

**4.1 Classification tree model accuracy**

The overall classification accuracy obtained with the CT model (Table 3) demonstrates that the employed biophysical drivers are strongly related to the FRs within the study area, most especially the percentage of area occupied by shrubland (SHR) and spring rainfall (RFAJ). Regarding FR-specific accuracy, however, results show a notable contrast between the first three FRs




(with a minimum accuracy of 69.9% and a maximum of 82%) and the fourth, with only 38.2% of all parishes correctly classified. This indicates that, although this FR possesses relevant distinctions in relation to the others from a wildfire-management standpoint (Fig. 3-B), it cannot be adequately discriminated using the set of biophysical drivers employed in this study. As the influence of social variables such as population or road density over FRs is well known (Pausas & Fernández-Muñoz, 2012; Rogers et al., 2020; Syphard et al., 2007), it is likely that their inclusion in the model would improve its accuracy, both in general terms and specifically in relation to FR4.


## 4.2 Relations between fire regimes and biophysical factors

### 4.2.1 FR1

FR1 is marked by the lowest burned area extension (CPAB) and the lowest wildfire frequency (AWWF) within the study area, with the resulting damage being relatively concentrated over time (GCI) (Fig. 3-B). Regarding its relation to the biophysical
drivers, it has the lowest values of percentage of parish area occupied by shrubland, the lowest amount of spring rainfall, the lowest slope inclination and the highest percentage of parish area occupied by agriculture. These are the four most important variables in the CT model, and their values in FR1 are in accordance with its low CPAB and AWWF. Shrubland abundance promotes extensive and frequent fires due to this land cover's well-known fire-proneness and quick regeneration (Bergonse et al., 2022; Moreira et al., 2009; M. C. S. Nunes et al., 2005; Oliveira et al., 2014, 2020). Meneses et al. (2018) focused on the
relation between LULC and probability of wildfire recurrence, associating shrubland to the highest probability values. Beside its inherent fire-proneness, there may also be a human factor promoting the burning of this LULC class as, due to its low monetary value, it is typically given a low order of priority in wildfire suppression strategies (Moreira et al., 2009, 2011). It is therefore expectable that FR1's low shrubland value will contribute to its low CPAB and AWWF. Spring rainfall can be assumed to promote vegetation growth and thus fuel availability during the summer months. This is in accordance with the
results obtained by Bergonse et al. (2022), who determined RFAJ to have a positive influence both over AWWF and CPAB. The positive effect of spring rainfall over annual burned areas has also been highlighted by different authors in the literature (Oliveira et al., 2012; Pereira et al., 2005; Xystrakis et al., 2014). Low spring rainfall values will, conversely, be associated with both low wildfire extensiveness and frequency. Slope inclination promotes wildfire spread (Carmo et al., 2011; Marques et al., 2011; Oliveira et al., 2020; Parente & Pereira, 2016), with the lowest slope values in the study area being in accordance
with the minimum CPAB values shown by FR1 in relation to all other FRs. Finally, agriculture's low fire-proneness is well known (Meneses et al., 2018; Moreira et al., 2009; Oliveira et al., 2014). Its relative importance in the parishes associated with FR1 will therefore contribute to their low CPAB and AWWF.



### 4.2.2 FR3

It seems appropriate to follow this discussion with FR3, as this FR possesses the exact opposite characteristics to FR1. It has the highest CPAB and the highest AWWF, along with the lowest GCI. Accordingly, it has the maximum percentage of shrubland and the maximum values of spring rainfall, as well as second highest slope values. FR3 is characterized by the lowest percentage of eucalyptus among all four FRs, which would seem contradictory given the relative fire-proneness of this LULC (Meneses et al., 2018; Oliveira et al., 2020; Xanthopoulos et al., 2012). However, this suggests that the fuel availability

behind FR3's high CPAB is mostly dependent on shrubland and its faster response to spring rainfall. This is confirmed by FR3's low Net Productivity Ratio, the lowest among all FRs, which is indicative of a relatively reduced forest cover. Nevertheless, FR3 is marked by the highest incidence of forests of broadleaves other than holm oak, cork oak and eucalyptus of all FRs. As this LULC class has a positive effect over CPAB and AWWF (Bergonse et al., 2022), this suggests that forest-type fuels also have some importance in FR3.

It is noteworthy that FR3 has the greatest burned area extension (CPAB), despite having the highest degree of forest patch fragmentation of all FRs. Although this variable was calculated in this work using only forest patches, it was previously shown by Bergonse et al. (2022; see Table S.1) to be strongly correlated to the fragmentation of shrubland and forest when considered altogether. It can therefore be interpreted as describing general fuel patch fragmentation. Although a high level of fragmentation can be expected to constrain extensive wildfires (Gralewicz et al., 2012; Ryu et al., 2007), the dominant fuel

type in FR3 is shrubland. Therefore, even if each individual wildfire is constrained in its spread, the quick regeneration of fuels will nonetheless allow for frequent burning, leading to an important accumulation of burned area over time.

### 4.2.3 FR2

FR2 can be considered to occupy an intermediate position between FRs 1 and 3 with respect to the three fire regime parameters considered (Fig. 3-B). It burns both more extensively and more frequently than FR1, but with a lower temporal concentration of damage. Its values in the different biophysical factors are mostly in accordance with its intermediate character.

### 4.2.4 FR4

FR4 is characterized by the second highest extension of burned area (CPAB). However, unlike FR3, this is accompanied by a

relatively low wildfire frequency (AWWF), leading to a relatively high temporal concentration of damage (GCI). Its elevated CPAB is in accordance with its values in several of the already discussed biophysical drivers: a relatively high percentage of shrubland, high spring rainfall, the highest slope inclination, and the lowest percentage of agriculture. FR4 also has a relatively high percentage of eucalyptus forests. The importance of forest cover in FR4 is indicated by the relatively high Net Productivity Ratio, which suggests that, unlike FR3, FR4 is more dependent on slowly regenerating forests as fuel instead of shrubland.





Significantly, FR4 also has the lowest degree of forest patch fragmentation among the four FRs. Together with the slow
        regeneration of forest cover, this would contribute to its relatively low AWWF and high GCI (Fig. 3-B), marking this FR as
        being dominated by infrequent, extensive forest wildfires.

        It is noteworthy that FR4 and FR2 share similar values with respect to the two most important biophysical factors in the CT
        model, namely shrubland and spring rainfall. However, FR4's higher slope, more extensive eucalyptus forests, less extensive

agricultural areas and lower degree of forest patch fragmentation justify its higher burned area extension, lower wildfire
        frequency and higher temporal concentration of damage.

        Notably, FR4 has similar percentages of area occupied by eucalyptus forests to FR1, which has the lowest CPAB of all four
        FRs. The fact that this is a relatively fire-prone LULC (Meneses et al., 2018; Oliveira et al., 2020; Xanthopoulos et al., 2012)
        suggests that, in our study area, the effect of eucalyptus's fire-proneness on CPAB is modulated by other factors, which hinder

its burning in areas associated with FR1, but not in those associated with FR4. Possible explanations would be FR1's relatively
        higher level of fuel patch fragmentation and the denser urbanization and increased human presence along the coast (where FR
        1 is concentrated), constraining fire spread and promoting a more rapid and efficient response in case of ignition.

**4.2.5 Biophysical factors with uncertain roles in the CT model**

Despite contributing to the CT model, the role of the least important variables in influencing the FRs in the study area is
        unclear. Regarding summer temperature (TPJS; Fig. 6-I), and assuming a homogeneous fuel distribution throughout the study
        area, it would be expected that higher values would promote fire-proneness, and therefore more extensive and/or frequent
        burning (Viegas, 2006; Viegas et al., 2004). However, the FR with the highest CPAB, FR3, has the lowest summer temperature,
        whereas that with the second highest CPAB (FR4) has a similar value to the FR with lowest CPAB (FR1). There seems to be

two possible explanations for these results. Firstly, the contrast in summer temperature between the FRs in the study area is
        relatively modest (in comparison with variables such as the percentage of shrubland or of agriculture, which have stronger
        differences). The difference between the highest and lowest medians (FR4 with 20.5°C and FR3 with 19.9°C) is only 0.6 °C.
        Such temperature differences may be insufficient to distinguish significantly between FRs. Secondly, fuel distribution within
        the study area is not homogeneous, as shown by the differences in the LULC variables among the FRs (Fig. 6). It is therefore

possible that the potential effects of summer temperature in burned area are constrained by fuel availability. In this regard,
        other authors have pointed out that the dependence of area burned on dry climatic conditions occurred only when fuel was not
        the main limiting factor (Pausas & Fernández-Muñoz, 2012).

        The final three variables in the CT model have only a very minor importance (Fig. 5). Invasive species (INV) (Fig. 6-J), forests
        of holm oak and cork oak (OAK) (Fig. 6-K) and forests of coniferous species other than maritime pine and stone pine (CON)

(Fig. 6-L) all are characterized by a predominance of very small values among the studied parishes, together with a high level
        of dispersion. Their values in the different FRs do not suggest clear patterns.



### 4.3 Implications to wildfire management

Among the FRs identified, those whose characteristics are likely to bring more challenges from a wildfire management
standpoint are FRs 3 and 4 (encompassing 188 parishes). These have the highest tendency to burn extensively over time, and
thus the highest potentials for material and human damage.

FR3 is characterized by frequent wildfires, without important contrasts in burned area from year to year, leading to gradual
and ultimately high accumulation of burned area over time. Fuel type and availability play a major role, with spring rainfall-
fed shrubland allowing for frequent burning. It is unlikely that individual wildfires are very extensive, as this area is marked
by the highest degree of LULC patch fragmentation of all. From a fire management perspective, priorities seem to be:

(a) reducing fuel availability through land use planning policies promoting shrubland removal or substitution by less fire prone
LULC types (such as agriculture or different types of forest) by landowners.

(b) reducing ignitions through sensibilization campaigns or legal constraints to the use of fire in critical areas and times of the
year.

(c) focusing existing early detection and combat capabilities on extinguishing the frequent wildfires at the earliest possible
stage. This implies the capacity for active combat in possibly several locations at the same time.

FR4 (102 parishes) is characterized by extensive burning over a relatively small number of wildfires, leading to a high temporal
concentration of damage. The low wildfire frequency, together with the relatively high net primary productivity (NPP) and the
importance of eucalyptus forests indicate that forest-type fuels are the most relevant to this fire regime's properties.
Nevertheless, shrubland is present and promoted by the relatively abundant spring rainfall. The extensive wildfires of FR4 are
promoted by the steepest slopes of all four FRs. Also, the lowest degree of forest patch fragmentation and the lowest percentage
of agricultural areas among all FRs suggest that fuel continuity and low human presence for early detection may also play a
role in defining the properties of this FR. Policy-wise, priorities seem to be:

(a) landscape management strategies, constraining fuel continuity, with possible measures including the implementation of
fuel breaks and promoting patches of less fire prone LULC types throughout the forested areas.

(b) focusing early detection and early response capabilities on ensuring that ignitions, although relatively infrequent, are not
allowed to develop. This implies quick mobilization of means to an ignition location and the timely allocation of surveillance
resources to the more hazardous areas, at the beginning of the main fire season.

The remaining FRs (1 and 2), which include most of the studied parishes (450 and 299, respectively) seem to show less
challenging conditions with regard to the priority for possible policy measures. FR2 has similar properties to FR3, although in
a lesser degree (less extensive and less frequent burning, greater temporal concentration of damage). FR1 has the least
extensive burned area and the least frequent wildfires of all FRs.

It is important to note that we have not taken aspects such as the exposure and vulnerability of infrastructure and populations
into account in this work. It is therefore likely that some wildfires in the coastal, highly populated parishes of FR1 will result
in more relevant damage than some of the wildfires in the sparsely populated interior areas of FRs 3 and 4. Our emphasis was



on the definition of fire regimes and on assessing their relations to a set of potential biophysical drivers, mainly related to fuel conditions that can be directly modified by human intervention. Future approaches to these topics should include the quantification of the damages associated with each fire regime within the study area (human losses, infrastructure, and the monetary values of different LULC patches), and the inclusion of social variables into the set of potential fire regime drivers.


## 4 Conclusions

Four distinct fire regimes can be differentiated among the parishes of the study area, based on their tendencies to burn extensively, to burn frequently, and for burned area damage to be concentrated over time. The first fire regime is marked by the least extensive burned area and the lowest wildfire frequency, as well as the maximum temporal concentration of damage.

The second fire regime is marked by more extensive burned areas and more frequent wildfires than the first, as well as lower temporal concentration of damage. The third and fourth fire regimes are characterized by the most extensive burned areas, and contrast in terms of wildfire frequency and temporal concentration of damage. The third fire regime has the most extensive burned area of all four fire regimes, as well as the most frequent wildfires, with burn damage dispersed through time. In contrast, the fourth fire regime has slightly less extensive burned areas, but a much lower wildfire frequency, with the damage

being more concentrated in time.

A classification tree model was used to relate the fire regimes to a set of 12 potential biophysical drivers. Results show that LULC, slope and spring rainfall are the most important drivers of the four fire regimes. The most relevant LULC classes are shrubland/spontaneous herbaceous vegetation, which is the foremost of all drivers, and agriculture, the first due to its fire-proneness and quick regeneration, and the second due to its constraints over wildfire spread. Slope exerts its effect by

promoting wildfire spread, whereas spring rainfall is a factor of fuel availability later in the year. Despite the good discriminating capacity of the classification tree model, other drivers, likely of a social nature, also influence the fire regimes in the study area. The model also showed unequal capacity to identify each the four fire regimes, with a markedly inferior accuracy in the case of the fourth.

The specificities shown by the two fire regimes marked by more extensive burned areas suggest different policies regarding

wildfire prevention and combat, with the foremost issues being fuel abundance and ignition frequency in one case, and fuel continuity in the other.

Our results highlight the fact that contrasting fire regimes may occur in close spatial proximity. Ignoring this can lead to errors both in terms of spatial planning policies and fire suppression strategies, such as disregarding the concentration of means of suppression where the damages are likely to be more extensive. By allowing to identify fire regimes in an objective and

reproducible way, the proposed methodological approach can be applied in other studies and other spatial contexts.



**Data availability**

The used dataset is available at the Zenodo repository: https://doi.org/10.5281/zenodo.6552804

**Acknowledgments**

This work was financed by national funds through FCT—Portuguese Foundation for Science and Technology, I.P., under the framework of the project "People&Fire: Reducing Risk, Living with Risk" (PCIF/AGT/0136/ 2017), under the programme of

'Stimulus of Scientific Employment— Individual Support' (contract 2020.03873.CEECIND), and by the Research Unit UIDB/00295/2020 and UIDP/00295/2020.

**Author contribution**

Rafaello Bergonse: Conceptualization, Writing – Original Draft, Writing – Review & Editing, Formal Analysis, Methodology.

Sandra Oliveira: Conceptualization, Methodology, Writing – Review & Editing. José Luís Zêzere: Writing – Review & Editing, Supervision. Francisco Moreira: Methodology, Writing – Review & Editing. Paulo Flores Ribeiro: Conceptualization, Methodology, Writing – Review & Editing. Miguel Leal: Writing – Review & Editing. José Manuel Lima e Santos: Conceptualization, Methodology, Project Administration

**Declaration of competing interest**

The authors declare that they have no conflict of interest.

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
