# Peer review of "Differentiating fire regimes and their biophysical drivers in Central Portugal"

_EGUsphere, 2022_

## Author Comment (AC3)

**Referee comment:**

**https://editor.copernicus.org/index.php?\_mdl=msover\_md&\_jrl=778&\_lcm=oc158lcm159n &\_ms=103240&salt=240954201780718781**

This manuscript aims to characterize fire regimes in Central Portugal and investigate the degree to which the differences between regimes are influenced by a set of biophysical drivers. The authors used civil parishes as units of analysis and cumulative percentage of parish area burned, Gini concentration index of burned area over time, and area-weighted total number of wildfires over a reference period of 44 years (1975-2018). The authors used cluster analysis to aggregate parishes into groups with similar fire regime and a classification tree model to assess the capacity of a set of potential biophysical drivers to discriminate between the different parish groups. The methods used seem to be suitable and the manuscript is nicely written. However, I have some criticisms, including some changes concerning the datasets used and the novelty of the present work, that should be addressed before considering the paper for publication in Natural Hazards and Earth System Sciences journal.

**Major suggestions/comments**

We thank the reviewer for the overall positive feedback and the pertinent suggestions provided. We have numbered each point and respond to it in detail below.

**1) Novelty of the work:**

The results are fairly described and the discussion focused in a reduced number of previous publications, including two previous works of the same authors that exhibits strong similarities with the present work. The novelty (and need) of the present results of is not clearly addressed. The baseline of the present work in terms of data and methods is very similar to the previous two works. The data used is the same and the statistical approaches are slightly different, but very related with the previous ones. The main results are the same: the role played by LULC, slope and spring rainfall in fire behavior.

The present paper adds the classification in 4 FR for central Portugal. However, the FR classification is closely dependent of the data used. This leads to my following comment.

**R:** Regarding the novelty of the work, the present manuscript is one of the results of a larger study, which includes Bergonse et al. (2022). Both works share a common rationale in that they share the same spatial analysis units and study area, the same three fire regime descriptors, and the same biophysical drivers. However, they have different objectives and analysis techniques. In Bergonse et al. (2022), relations between the biophysical drivers and each of the three fire regime descriptors were separately analysed using ordinal regression equations. Although the study area was assumed to have a single fire regime, the spatial patterns shown by the three fire regime descriptors suggested the existence of distinct regimes.

The present work builds upon the previous results, in that we employ cluster analysis to explicitly identify and then characterize the different fire regimes within the study area, something which has not been done before. We subsequently apply a classification tree model to assess the capacity of the different biophysical drivers to discriminate between the four fire regimes

defined. After interpreting the results, we then discuss the implications of the identified fire regimes and their drivers to wildfire management. This also is a completely novel outcome.

Being results of a common, ongoing research project, this manuscript and the previous article can be considered complementary. Each of these works, however, presents distinct and novel results, which is why we feel this manuscript to the suitable for publication in this journal. We believe that the findings presented are valuable and helpful towards understanding the complexity of wildfires in Portugal as each small scientific step adds to the knowledge we very much need to deal with such issues.

In the manuscript, the relations between this and the previous work are made explicit in the final part of the Introduction (lines 57 and following).

We would also like to note that the other work mentioned by the reviewer (Oliveira and Zêzere, 2020) was not a part of the research project mentioned above, having different study area, temporal scope, and analysis technique (random forest). It also employs as dependent variable only one of the three fire regime descriptors mentioned above. In fact, this paper investigated the relation between the spatial distribution of burned area and different biophysical and social drivers for the parishes of the whole mainland Portugal, in a rather different scope than the one now presented.

**2) Datasets:**

FR regime classification in strongly dependent of the historical data over the region. Therefore, the use of climate data than does not describe the last two decades, when we are facing a change in fire paradigm over Europe, with the occurrence of the so-called megafires, highlights the fragilities of the FR classification.

Besides the 'old' meteorological datasets, the higher fire intensity or severity of the observed fire behavior trends was not included in the FR classification. The authors used burned area, however the burned area inside a civil parish may not be a good indication or fire intensity; other parameters (available through remote sensing datasets) should be included.

The inclusion of the suggested datasets, considering the aim of the present work, will strongly improve the quality of the results, highlighting its novelty.

**R**: The issue of the temporal limitations of the climate data is considered in detail below, in comment 4. However, we would like to underline that, in accordance with the adopted fire regime definition (lines 33-34 of the manuscript), the fire regimes were described based on the consequences of fires in terms of burned area through time, and not on the meteorological context in which these fires take place. The characterization of fire regime is based on burned area data for the 44-year period between 1975 and 2018 and is therefore not subject to any fragility derived from climate data limitations. Climate data were subsequently used as possible biophysical factors influencing fire regime, and it is in relation to this aspect of the work that the temporal limitations of the climate data indeed constitute a fragility. This limitation is acknowledged in the Data Collection and Pre-Processing section (lines 161-164) and will be

further highlighted in a subsection to be included in the Results and Discussion section, focusing on the uncertainties and limitations of this work.

On a sidenote regarding the properties of the fire regime and their possible change between the period encompassed by the climatic data (1975-2000) and the later years, the following experiment was made. We created a new variable describing the cumulative percentage of parish area burned (CPAB) between 1975-2000 and ranked all study parishes by their value in this variable. We then ranked all parishes as to their value in the CPAB used in the article (i.e., encompassing the whole study period 1975-2018). We then calculated the Pearson correlation coefficient between the two ranked variables, obtaining an *R* of 0.895, significant at the 0.01 level. This result shows that the relative positions of the different parishes in terms of cumulative percentage of area burned are quite similar, regardless of the period considered. Although we tried this only for this fire regime descriptor, this result strongly suggests that the fire regimes among the studied parishes show a similar behaviour whether we limit the analysis to the climate-data covered period or to the whole 44-year period used in the article.

Regarding the issue of wildfire intensity/severity, fire regimes can be described with greatly varying degrees of complexity. We purposefully employed a simple, straightforward approach, expressing it with three indicators that can be extracted from freely available annual burned area maps, and therefore easily reproduced in other study areas. We agree that severity is an important aspect of fire behaviour that may not be adequately expressed by burned area alone, as are others such as the characteristics of the largest, relatively infrequent fires (which would include the so-called megafires). We will refer to these aspects of fire regime in a future "Uncertainties and Limitations" subsection of the Results and Discussion section, to inform future studies. We believe, however, that the focus on a simpler approach and the absence of these other datasets does not hinder the usefulness or novelty of the work, considering its objectives.

**3) Slope:**

"Topography was expressed by slope (80th percentile, in degrees), which can be expected to promote flame propagation". Why using the 80th percentile and not 90th or 75th. Did you make a sensitivity analysis for this choice? Did the authors include elevation information? Why?

**R:** The set of 12 biophysical variables employed are derived from our previous results in Bergonse et al. (2022), as stated in lines 109-110 of the manuscript. In the referenced article, we initially adopted both slope and elevation as potential biophysical controls, using percentiles 50, 75, 80, 90 and 95. During a multicollinearity analysis, all percentiles were shown to be strongly intercorrelated. We thus chose to keep those more strongly correlated with the remaining percentiles in the same group, leading to the adoption of the 80th percentiles of slope and altitude. Altitude was eliminated further along the multicollinearity analysis process, as its Variance Inflating Factor showed it can be expressed as a linear combination of other variables in the dataset. This process is described in the Data Analysis section of Bergonse et al. (2022).

**4) Rainfall:**

"RFAJ was calculated from monthly rainfall data obtained from the Worldclim database (1970-2000)". The authors present an assessment for 44 years (1975-2018) and one of the crucial datasets used is only characterizing half of the period. The data used is not representative of the fire regimes in Portugal, namely considering the fire behavior in the XXI century. Please use an alternative database for precipitation data that characterize the entire period evaluated, e.g. ERA5 (1979 to present), or alternative change the period of analysis to 1975-2000.

..."in the form of raster maps of approximately 30 seconds (about 1 km resolution), which were resampled to a 25 m pixel". How was done the resampling? Which co-variates were used to do resampling? And, why to do the downscale if the data is further aggregated at parish level?

**Temperature:**

The data used is not representative of the fire regimes in Portugal, namely considering the fire behavior in the XXI century. Please use an alternative database for temperature data that characterize the entire period evaluated, e.g. ERA5 (1979 to present), or alternative change the period of analysis to 1975-2000.

**R**: We will focus on the issue of the climate dataset first, and then respond to the comment on the resampling process.

In the manuscript, we acknowledge the disparities between the periods used to characterize the fire regimes and the biophysical drivers in lines 161-165 and in Table 1. We thank the reviewer for the suggestion to use the ERA5 dataset to overcome this issue, which we have investigated. Unfortunately, the spatial resolution of ERA5 makes it too coarse to be applicable to a study on such a detailed scale as the one we employ. ERA5 is made available with a 0.25-degree resolution, which translates to a pixel of approximately 24.8 km. An overlay between an ERA5-derived map and the limits of our study parishes shows that each pixel comprises multiple complete parishes within it, making the ERA5 dataset too generalized for application in this study. In comparison, the Worldclim dataset used has a resolution of approximately 1000 m, which makes it suitable for our scale of analysis. A solution to this issue would be to limit the study to the period 1975-2000, as suggested. However, this would entail a similar problem with the land-use data, which only begins in 1990 (1995 in the cases of two specific variables) (as shown in Table 1).

The use of these imperfectly overlapping datasets, imposed by the unavailability of suitable data, implies the assumption that all are representative of the long-term, general fire regimes and biophysical factors that have characterized the study area within the last four decades. We agree that record fires, such as those seen recently, are important to understand the dynamics of fire in different contexts. However, they are not so important in relation to our approach, as they would detract from the general tendencies of the fire regimes we wish to characterize, moreover when our approach is based on annual burned area data and not on the characteristics of individual fires.

The resampling of the climate data maps was performed using ArcMAP's *Resample* tool, using nearest neighbour assignment. This software was used for all spatial analysis operations, as stated in lines 166-7 of the manuscript. The resampling was done to minimize generalization in association to the Zonal Statistics tool used to calculate the mean values for the pixels in each parish. The following example can clarify the rationale behind the resampling. Let us imagine that our purpose is to calculate the mean temperature during the summer months for each of a

set of parishes. To do so, we will use the vector map with the parish limits, and a raster map with the variable of interest, that is, temperature during the summer months. Let us assume this raster map has 1000-m pixels. If one of the parish polygons partially overlays a 1000-m pixel without covering its centre, the value of this pixel will be ignored in the calculations, i.e., the mean value calculated for that parish will in practice ignore a part of the surface of the parish. If, however, the raster map has a 25-m pixel, a much smaller part of each parish will suffer from this problem, as each polygon will encompass a much greater number of smaller cells, which will fill its area much more fully.

**5) Net Primary Productivity (NPP)**

Please consider to use the most recent version of the NPP product (2000-Present). Alternatively, consider to use the Climate Data Record of NDVI (annual mean or sum) available since 1981 to present (https://www.ncei.noaa.gov/access/metadata/landing-page/bin/iso?id=gov.noaa.ncdc:C01558).

**R:** The newer version of the MODIS NPP dataset (061) is indeed more up to date, as it extends to the present day. However, regarding our study period (1970-2018, the period used to characterize the fire regimes) the newer MODIS dataset includes only four more years (2015-2018), which are unlikely to alter the general, long-term tendencies in which this article is focused.

Regarding the NDVI dataset from NOAA, we thank the reviewer for the suggestion, which we investigated. The data is provided with a resolution of 0.05 degrees. This translates to a pixel of approximately 2946 m, which is much coarser than the 500-m NPP dataset we employed, even though it could still fit with our analysis. There are always options to make when it comes to the data integrated; in this case, we believe the NPP data is a reasonable choice considering the long-term approach applied, which does not require daily data. We will, nevertheless, make reference to the NOAA dataset in a future *Uncertainties and Limitations* subsection, to be inserted into the *Results and Discussion* section. This will inform future studies and research projects.

**6) Lines 152-160**

This paragraph seems to be out of order. Please consider to reorganize the paragraph (before NPP paragraph).

**R:** The paragraph will be repositioned as suggested.

**7) Lines 161-165**

The different periods considered for the different datasets could have a strong impact on the results, as the spatial patterns for precipitation and temperature in the last 30 years of the XX century may have strong differences in compared with NPP in the first 20 years of the XXI century.

With the aim to have a fire regime description that really reflects the recent vegetation, climate and fire behavior trends, I strongly suggest to include: a) Temp and Precip from ERA5 from 1979-Present; b) NDVI from 1981-Present. Therefore, the period of analysis would be 1981-Present (41 years).

**R:** We thank the reviewer for the suggestions. The issues regarding the climate dataset and the NPP dataset are considered, respectively, in the responses to comments 4 and 5.

**8) Lines 360-361**

"This is confirmed by FR3's low Net Productivity Ratio, ..., which is indicative of a relatively reduced forest cover." Please provide a reference or provide the analysis that allow this statement (or remove the sentence)

**R:** In our previous work, it was observed that NPP was inversely correlated to the percentage of the area of each parish covered with shrubland, but positively correlated to the percentage of eucalyptus forests, pine forests and forests of invasive species (please see Bergonse et al., 2022, supplementary table S.1). We will reference this work at this point.

**9) 4.2.3 FR2**

Please provide a better characterization of this FR. As it is, seems that this fire regime is not a separate fire regime and may indicate that the classification in 4 fire regimes is not the adequate

**R:** Throughout section 3.1, our choices regarding the clustering process are described and justified. Figure 2 shows that a 3-cluster solution would indeed express, in a more synthetic way, the major contrasts in fire regime across the study area (as stated in lines 262-263). As we also note, cluster 2 always clearly occupies an intermediate position, regardless of the choice between 3 and 4 clusters (Figure 3 and lines 250 and 255-7). We then justify the option for a four-cluster/fire regime solution based on the practical implications of the fourth cluster (a subset of what was cluster 2 in the 3-cluster solution) regarding wildfire prevention and suppression (as stated in lines 263-266).

We understand the issue raised by the reviewer, in that, when reading the discussion of each cluster in section 4.2, a reader may at first not see the justification for keeping cluster 2, a cluster that has apparently no distinguishing features. However, we feel that such a doubt could only arise from a partial reading of the article, not only because cluster 2 is present in both 3- and 4- cluster solutions, being in either case relevant regarding fire regime variability in the study area (as shown in Figure 2), but also because it possesses intermediate characteristics in both clustering solutions. Indeed, within our study area, this cluster/fire regime it distinguished by this intermediate character.

**10) Lines 401-408**

The less clear relation with summer temperature is probably related with less adequate database used, that does not reflect the temperature changes in the last two decades. Please check the impact of use of the suggested dataset for meteorological parameters.

**R:** We considered the issue regarding the climate dataset in our response to comment 4. Although we consider different possible explanations for the results obtained (lines 401-412), we agree with the reviewer in that there may exist a relation between the limitations in temporal extent of the climate dataset and our results. This will be made explicit in the "Uncertainties and Limitations" section that will be incorporated into the Results and Discussion section.

**11) Lines 409-412**

Is the statement supported by the NPP results of this work? Please explain.

**R:** The distribution of NPP values among the four FRs (Figure 6-G) shows that the lowest value is associated to FR3, which has the highest Cumulative Percentage of Area Burned (CPAB) of all FRs. This low NPP is in agreement with the highest percentage of area covered by fire-prone, quickly regenerating shrubland, also shown by FR3 (Fig. 6-A). This would support the notion that fuel is the factor controlling burned area in this FR, instead of summer temperature (which has its lowest value in FR3).

Summer temperature has its highest values in FRs 1 and 4. The highest NPP values also occur in these two FRs, in accordance with the highest percentages of area covered by eucalyptus forest, also shown by these two FRs (Fig.6-E). These values, suggesting the combination of available forest fuels and high summer temperatures, only translate into extensive burned area in the case of FR4, which has the second highest CPAB of all FRs (Fig.3-B). In the case of FR1 (with the lowest CPAB of all FRs), factors other than fuel availability seem to constrain burned area. Among the biophysical drivers considered in this work, examples could be the high percentage of agricultural area (fragmenting fuels and implying higher human presence and therefore quicker detection and response) (Fig.6-D), the lowest slope of all four FRs (slowing flame propagation; notably, FR4 has the highest slope values) (Fig. 6-C), and the relatively high fragmentation of fuel (in comparison to FR4) (Fig. 6-H). Additionally, FR1 occurs mostly along the coastal sector, which has higher rates of urbanization and population densities than the inland parishes. It is likely that these will promote faster detection and better response capabilities on the part of the authorities in case of ignition.

In sum, NPP results seem to support the idea that fuel availability constrains to a degree the effect of summer temperatures. However, the role of NPP must be considered together with the effects of other factors, as fire regimes result from the interaction of multiple drivers.

Minor

12) Lines 90-95

Changed format.

**R:** The paragraph will be correctly re-formatted.